# Passive Surveillance of Human-Biting *Ixodes scapularis* Ticks in Massachusetts from 2015–2019

**DOI:** 10.3390/ijerph20054306

**Published:** 2023-02-28

**Authors:** Alexandra Sack, Elena N. Naumova, Lori Lyn Price, Guang Xu, Stephen M. Rich

**Affiliations:** 1Clinical and Translational Science Graduate Program, Tufts University Graduate School of Biomedical Sciences, Boston, MA 02111, USA; 2Department of Biological Sciences, Eck Institute of Global Health, University of Notre Dame, Notre Dame, IN 46556, USA; 3Division of Nutrition Epidemiology and Data Science, Friedman School of Nutrition Science and Policy, Tufts University, Boston, MA 02111, USA; 4Tufts Institute of Clinical and Translational Science, Tufts University, Boston, MA 02111, USA; 5Institute of Clinical Research and Health Policy Studies, Tufts Medical Center, Boston, MA 02111, USA; 6Laboratory of Medical Zoology, Department of Microbiology, University of Massachusetts, Amherst, MA 01003, USA

**Keywords:** *Ixodes scapularis*, tick-borne diseases, *Borrelia burgdorferi*, *Anaplasma phagocytophilum*, *Babesia microti*, *Borrelia miyamotoi*

## Abstract

This study aimed to analyze human-biting *Ixodes scapularis* ticks submitted to TickReport tick testing service from 2015–2019 in Massachusetts to (1) examine possible patterns of pathogen-positive adult and nymphal ticks over time and (2) explore how socioeconomic factors can influence tick submissions. A passive surveillance data set of ticks and tick-borne pathogens was conducted over 5 years (2015–2019) in Massachusetts. The percentages of four tick-borne pathogens: *Borrelia burgdorferi, Anaplasma phagocytophilum, Babesia microti*, and *Borrelia miyamotoi* were determined by Massachusetts county and by month and year. Regression models were used to examine the association between zip-code-level socioeconomic factors and submissions. A total of 13,598 *I. scapularis* ticks were submitted to TickReport from Massachusetts residents. The infection rate of *B. burgdorferi, A. phagocytophilum*, and *B. microti* was 39%, 8%, and 7% in adult ticks; 23%, 6%, and 5% in nymphal ticks, respectively. A relatively higher level of education was associated with high tick submission. Passive surveillance of human-biting ticks and associated pathogens is important for monitoring tick-borne diseases, detecting areas with potentially high risks, and providing public information. Socioeconomic factors should be considered to produce more generalizable passive surveillance data and to target potentially underserved areas.

## 1. Introduction

The causative pathogens of tick-borne diseases have been associated with increased tick density in *Ixodes* spp. ticks in multiple states in the United States [1,2]. Increased *Ixodes* spp. tick density has been associated with oak forests, higher humidity, and denser litter cover [1,3,4]; however, these studies involve collecting ticks from the environment and do not account for human exposure to ticks. Environmental factors can be considered to fall into two categories: those that affect tick mortality and a tick’s life cycle, and those that affect questing behavior [5]. For humans to be infected by a tick-borne pathogen, human land use and pathogen-positive ticks must spatially overlap [6]. Human behavior also plays an important role in determining access and land use, influencing tick bites and pathogen exposure [7,8]. Field collection of host-seeking ticks usually provides vector density and geographic locations of ticks. However, as human behavior also strongly affects risk, three surveillance methods, (1) human disease cases, (2) serology of domestic animals as sentinels, and (3) human-biting ticks, can help fill in knowledge gaps by accounting for factors associated with human behavior. [9,10,11,12]. Lyme disease patients often are unaware of a tick bite preceding the onset of symptoms; moreover, sentinel serology and human case reports relying on the place of residence may fail to account for travel history and may misattribute a high percentage of cases [13]. Our current paper not only examines the exposure risk from human-biting ticks but also explores the effect of socioeconomic factors on passive surveillance.

Previous studies of human-biting *Ixodes scapularis* include those from Canada [12,14,15,16], single states in the United States [17,18,19,20,21,22,23], and two multistate studies [24,25]. These studies identified tick species and their life stages, the peak season of human-biting occurs, tick bite sites on the human body, and the prevalence of tick-borne pathogens. The human-biting tick data are important to public health, for it predicts spatial and inter-annual patterns of tick-borne disease case incidence. Passive surveillance provides important information about human-biting ticks and information on actual encounters with ticks and tick-borne pathogens. Analyzing the submission patterns for human-biting ticks provides insight into potential high-risk areas or groups to target with future public health surveillance programs. 

Massachusetts is a high-risk area for tick-borne diseases, with the most common diseases being Lyme Disease, Babesiosis, and Anaplasmosis [26,27,28]. *Borrelia burgdorferi* and/or *Borrelia mayonii* are causative pathogens of Lyme disease in humans, *Anaplasma phagocytophilum* causes Anaplasmosis, and *Babesia microti* causes Babesiosis. *Borrelia miyamotoi* causes *Borrelia miyamotoi* disease, an emerging disease in Massachusetts. All four of these diseases are spread by *I. scapularis* [26]. Strong seasonality with a peak in summer is recorded throughout New England in *I. scapularis* [18,29,30]; however not all tick-borne diseases carried by *I. scapularis* peak at the same time. *Borrelia miyamotoi* disease in humans occurs most commonly in July and August after the peak season for Lyme disease, which is in June and July [31]. Massachusetts currently performs tick exposure and syndromic disease surveillance but does not report diseases specific case numbers. Still, more than 0.2% of all emergency room visits were related to tick-borne diseases during the summer [32]; however, this also leaves a gap in current information about tick-borne disease exposure risks in the state. 

TickReport is a public outreach service at the University of Massachusetts at Amherst, providing individuals with information about potential pathogen exposures associated with tick bites. For the first several years of its existence, TickReport was small and served mostly communities close to the campus. As the service grew in popularity and appeal, the sampling density became more substantial such that this individual risk assessment service grew to comprise a passive surveillance network in aggregate [18]. In this paper, we analyzed human-biting *I. scapularis* ticks submitted to TickReport tick testing service from 2015–2019 in Massachusetts to (1) examine patterns of pathogen-positive adult and nymphal ticks over time and (2) explore how socioeconomic factors can influence tick submissions. 

## 2. Materials and Methods

### 2.1. Tick Collection, Identification, and Pathogen Detection

The ticks submitted to TickReport and employed for this study were submitted voluntarily to the TickReport from January 2015 through December 2019. All submitters were asked to provide information about the presumed exposure location, the tick removal date, and the person’s sex, age, and residence location. Each submission corresponded with a single tick and was treated as a separate exposure. This service is subjected to a fee and is available for the entire United States. While there was a variety of tick species submitted and states covered by this service, the present work was focused on *I. scapularis* ticks received from the Massachusetts area due to having the most complete information temporally and geographically. Information about the biting tick’s species and transmitted pathogens were ascertained by an expert [13,18]. Ticks were first morphologically identified to stage and species levels [33,34,35], then confirmed by molecular assays targeting the tick mitochondrial 16S rRNA gene and ITS gene; see reference for a list of primers [13,18].

The total DNA was extracted from each tick using Epicenter Master Complete DNA and RNA Purification Kits (Epicenter Technologies, Madison, WI, USA) following the manufacturer’s protocols. *B. burgdorferi s. l.*, *B. miyamotoi*, *B. mayonii*, *B. microti*, and *A. phagocytophilum* were detected by a multiplex TaqMan real-time PCR assay in 16 μL reaction volumes using the Brilliant III qPCR Master Mix (Agilent, La Jolla, CA, USA) in an Agilent MX3000P qPCR System. Cycling conditions included an initial activation of the Taq DNA polymerase at 95 °C for 10 minutes, followed by 40 cycles of 95 °C for 15 seconds and 60 °C for 1 minute. *Borrelia* detection was performed by first applying a *Borrelia* genus-specific detection assay for a conserved target, followed by specific qPCR assays for each of the three species (*B. burgdorferi s. l.*, *B. miyamotoi*, and *B. mayonii*) [13,18]. In Massachusetts, the canonical *B. burgdorferi* species found in *I. scapularis* is *B. burgdorferi* stricto sensu, the sole species of Lyme Disease in North America [36].

### 2.2. Human-Biting Ticks and Prevalence of Pathogens

Only ticks with a location of exposure in Massachusetts were included. The percentage of pathogen-positive ticks and tick submissions by life stage were calculated by month and total submissions were calculated by month and year with 95% confidence intervals (95% CI). The total tick submissions and the percentage of each of the four tick-borne pathogens were calculated per zip code tabulation area (ZCTA) and by county with 95% confidence intervals. The percentages of pathogen-positive adult and nymphal ticks per ZCTA and county also were calculated for each year and the study period. The annual trend of pathogen prevalence was analyzed by the Mann-Kendall Test at *p* < 0.05 level. 

### 2.3. Socioeconomic Characteristics Associated with Human-Biting Tick Submissions by Massachusetts Residents

Only submissions with a valid Massachusetts ZCTA (from residences in ZCTAs within Massachusetts) were included for the socioeconomic analysis. Median household income, percentage of self-reporting race as white, percentage of the population with a high school education or less, and population density, were collated for each ZCTA from the 2018 American Community Survey (ACS, 5-year estimates), or from the 2014 ACS or corresponding census blocks if 2018 data were not available. Distance from the TickReport laboratory was calculated as a straight-line distance from the ZCTA centroid. The land-use type for each ZCTA was calculated from the 2010 National Land Cover data set (NLCD) and was defined as the land-use type that made up the highest percentage in that ZCTA. The NLCD 2010 has twenty different categories based on 30-meter squares. All spatial analyses were performed using ArcMap 10.7.1 (ESRI, Redlands, CA, USA, 2020). Statistical analyses were run using RStudio 1.2.502 (R Studio Team, Boston, MA, USA, 2020). 

It was decided a priori to test the socioeconomic factors with and without Boston, due to the large range of socioeconomic statuses but consistently low tick submissions. Population density and median household income were transformed using a log transformation due to non-normal distributions. A negative binomial regression model was applied to examine associations between ZCTA-level socioeconomic variables and the tick submissions per ZCTA in both univariate and multivariable models. Socioeconomic variables that met the inclusion criteria of *p* < 0.10 in the univariate analysis were included in the multivariable analysis. Results from the regression models were reported as incident rate ratios. The land-use category that made up the highest percentage of each ZCTA was added to the final socioeconomic model to analyze the effect that the inclusion of land use had on the socioeconomic variables. Nagelkerke’s pseudo-R^2^ was measured for the models with and without land use [37,38]. Influence points were defined as ZCTAs with either residual greater than the absolute value of three and/or Cook’s value greater than 0.2. 

## 3. Results

### 3.1. Human-Biting Ticks and Prevalence of Pathogens

A total of 13,598 *I. scapularis* ticks were submitted to TickReport with a reported Massachusetts exposure: 76.7% of ticks were adults (*n* = 10,435), 21.6% were nymphs (*n* = 2935) and 1.7% were larva (*n* = 228). Men (*n* = 6743) and women (*n* = 6701) submitted a similar number of ticks from exposures (*n* = 154 chose not to include gender). Of all submissions, 96.6% reported exposure location at the ZCTA level. Also, 77.4% of submissions were from exposures to a tick in the same ZCTA as where the person lived. Over the five-year period, no ticks were submitted from 11.1% (*n* = 59/537) of Massachusetts ZCTAs (Figure 1). Most counties submitted more adults than nymphs; however, Nantucket and Dukes County submitted more nymphs than adults (74.2% and 57.1% of submissions, respectively), as only 16 adult ticks were submitted from Nantucket. 

In adult ticks, 39.0% (95% CI: 38.1–39.9%) were positive for *B. burgdorferi*, 8.1% (95% CI: 7.6–8.6%) for *B. microti*, 7.6% (95% CI: 7.1–8.1%) for *A. phagocytophilum*, and 2.0% (95% CI: 1.7–2.3%) for *B. miyamotoi*. In nymphal ticks, 23.1% (95% CI: 21.6–24.7%) were positive for *B. burgdorferi*, 6.4% (95% CI: 5.6–7.4%) for *B. microti*, 4.9% (95% CI: 4.2–5.8%) for *A. phagocytophilum*, and 1.3% (95% CI: 0.9–1.8%) for *B. miyamotoi*. No ticks were positive for *B. mayonii*. For adult and nymphal stages combined, 41.5% (95% CI: 40.7–42.3%) of ticks were infected by one tick-borne pathogen; 8.8% (95% CI: 8.3–9.3%) of ticks were infected by more than one pathogen. Five adult ticks were positive for all four pathogens. The prevalence of each pathogen varied by year, and no pathogen exhibited a significant linear trend (Table 1). Two larva each were positive for *B. burgdorferi*, *A. phagocytophilum*, and *B. miyamotoi.*

Exposure to *I. scapularis* forms two peaks with the first in the spring and early summer, April–June, and the second in the fall, October–November. The start of the tick season and the proportion of ticks in each peak varied by year with 2018 having the highest summer peak and 2017 the highest fall peak proportionally. Two distinct peaks were seen for exposures to adult *I. scapularis* ticks with nymphal ticks showing a single peak in the late spring/early summer (Figure 2). Pathogen prevalence remained consistent throughout the year with a decrease in *B. burgdorferi* going into late fall. Prevalence can drop to zero in low submission months for both nymphs and adults. 

The average prevalence of each pathogen varied county by county, and *B. burgdorferi* had the highest prevalence in every county, except for nymphs in Hampden County (Table 2). Only Nantucket had greater than 50% of adult ticks test positive for *B. burgdorferi.* Adult ticks in Essex County tested the highest for *B. microti* and *B. miyamotoi*, and Berkshire County for *A. phagocytophilum.* For nymphs, Suffolk County tested the highest percent pathogen-positive for *B. burgdorferi,* Plymouth County for *B. microti* and *B. miyamotoi*, and Hampden County for *A. phagocytophilum*.

### 3.2. Socioeconomic Characteristics Associated with Human-Biting Tick Submissions

For the socioeconomic analysis, we excluded 61 ZCTAs in the Boston area, as decided a priori. All socioeconomic variables met the inclusion criteria of *p* < 0.10 for the multivariable regression model (Table 3). ZCTAs with a higher percentage of the population self-reporting white race were positively associated with the total tick submissions per ZCTA (Table 3). Higher ZCTA-level median household income and percentage of the ZCTA with a high school education or less were associated with fewer ticks submitted from a ZCTA. Longer distance from the submitter’s residence to the TickReport lab was positively associated with the number of tick submissions (IRR = 1.003 per 1 km: 95% CI 1.002–1.006). Once land-use categories were added to the model, the percentage of self-reporting race as white became non-significant but was still insignificantly positively associated with submissions. The estimate for percent of the ZCTA with high school education or less and median household income changed by less than 10%. In both models, with and without land use, the log of population density was non-significant. 

Based on model diagnostics, the one ZCTA (01002 Amherst, *n* = 620 ticks submitted) that met the criteria for influence points was removed, and the model was rerun (Table 3). After removing the potential influence point, the direction of the association between the socioeconomic variables and the number of submissions did not change. However, the percentage of self-reporting race as white became significant, even with land use included, and median household income became insignificant. When examining the socioeconomic model with the Boston area included without land-use categories (Table 3), the direction of the association between the socioeconomic variables and the number of submissions did not change. However, the estimate changed towards the null for all variables but population density, which was significantly associated with fewer tick submissions. 

## 4. Discussion

Passive surveillance of human-biting is an important surveillance strategy that provides insight into potential high-risk areas or groups. The percentage of *I. scapularis* positive for tick-borne pathogens in Massachusetts varied by location and life stage. *B. burgdorferi* was the most common pathogen, followed by *B. microti* and *A. phagocytophilum.* Furthermore, we found that tick submissions were more common from ZCTAs with a higher level of education, even after accounting for land use. 

### 4.1. Human-Biting Ticks and Prevalence of Pathogens

Among ticks with a known exposure location, 77% of people experienced tick exposure in the same ZCTA as where they lived. Having exposure location thus prevented geographical misattribution for almost 25% of tick submissions and allowed for percent pathogen-positive and exposure risks to be mapped correctly. Even using the same diagnostic service with a similar percentage of adult ticks (77% vs. 81% in the previous report), the percentage of all ticks positive for *B. burgdorferi* has increased to 35.0% from 29.6%, which was reported by TickReport from 2006–2013. *B. microti* saw an even larger percentage increase from 4.6% to 7.6% and *A. phagocytophilum* from 1.8% to 6.9% of all ticks submitted from exposure in Massachusetts [18]. This is especially important as Massachusetts is not considered part of the *I. scapularis* expanding range [39]. Even if the pathogen prevalence in all questing ticks is not changing, people may be encountering more pathogen-positive ticks due to land use and behavior patterns. 

The exposure pattern for human-biting *I. scapularis* ticks followed the two seasonal peaks that are seen in field studies in the study area [40,41]. Adult submissions form two peaks with the nymphs forming a single peak [41]. This also follows the pattern of tick exposure visits in Massachusetts, though not disease visits [32]. Since this data set relies on tick submissions, there was a potential for human factors to affect this pattern. More work is needed to look at the seasonality of pathogen prevalence and not just seasonality in human cases, which depends on tick numbers and human behavior as well. Some climate variables have previously been associated with the percentage of ticks positive for *B. burgdorferi*, so seasonal prevalence trends may be more dependent on climate and local weather data than overall monthly trends [42,43,44,45]. 

Massachusetts is known to be an area of high risk for Lyme disease, and a high percentage of ticks are pathogen-positive [27,28]. While not significant, the number of ticks did increase across the study period; this is most likely due to increased awareness of services and not necessarily an increase in tick exposures. Two studies that collected questing ticks from high-risk areas in Massachusetts found *B. burgdorferi* in more than 60% of *I. scapularis* ticks [27,28], which was also seen in Nantucket in this study, even with a low number of adult tick submissions. Historically, babesiosis was diagnosed in Nantucket, Martha’s Vineyard, and coastal Massachusetts, but recently other inland areas have also had human cases [46]. A previous study from 2006–2012 using the TickReport passive surveillance system found *B. microti* to be almost entirely limited to Cape Cod and the islands [18]. In our study, the highest percent positive for *B. microti* was in Cape Cod (Barnstable County) and Essex County; however, *B. microti* was found in ticks in all counties, except Suffolk County, which may be due to having the lowest tick submissions. These areas represent the expanding range of *B. microti*. TickReport data from 2006–2012 found *A. phagocytophilum* limited to the eastern half of the state, and the highest percent positive is still in eastern Massachusetts. Fewer studies have been performed on the range of *B. miyamotoi*. A 2010–2012 study of ticks collected from Cape Cod found that 2.8% of female, adult ticks were infected with *B. miyamotoi* [40], which is similar to the percentage of adult ticks in the current study. Risks for tick-borne diseases are even higher when considering all four pathogens, as almost 10% of human-biting *I. scapularis* were positive for multiple pathogens. 

In nymphs, the overall percent positive for any pathogen was lower, as expected for fewer blood meals. It has been suggested that human-biting nymphs are especially important for assessing disease risk to humans [47]. Due to their small size, nymphs tend to be attached longer than adults before removal or go unnoticed [48]. Nymphs are assumed to be the source of most human cases, especially when no tick exposure is reported [47]. Host-seeking infected nymphs tend to be clustered spatially at a smaller spatial scale than adults, so human land usage likely plays a large role in exposure to pathogen-positive nymphs [49]. 

### 4.2. Socioeconomic Characteristics Associated with Human-Biting Tick Submissions

There is a fee associated with submission and previous socioeconomic associations have been reported with human Lyme disease cases, emphasizing the importance of examining socioeconomic factors associated with tick submissions. Amherst residents submitted the most ticks of any locale, by a large margin, likely due to TickReport’s location at the University of Massachusetts, Amherst. ZCTAs with higher percentages of post-high school education levels were associated with increased tick submissions, as was a higher percentage of residents self-reporting race as white once Amherst was removed from the model. These socioeconomic variables remained significant even after accounting for land use, indicating it was more than just that certain socioeconomic statuses could live in areas with higher risk, such as more forested areas. Land use such as deciduous forests, edge forests, and non-agricultural land has been previously associated with tick numbers [50,51,52]. The inclusion of Boston resulting in population density being the most significant factor was expected due to the high population density compared to the rest of the state and low tick submissions.

Studies of Lyme disease cases in the United States have found that cases are associated with relatively higher proportions of the population reporting race as white, higher levels of education, and lower levels of poverty [53,54,55]. For Anaplasmosis, one study found a relatively higher proportion of race reported as white and lower unemployment were associated with more human cases [54]. This may reflect both risk for infection, the costs of medical care, and the awareness of tick-borne diseases that is needed to seek diagnosis and treatment. Our study examined median household income, and not poverty, which might explain why after adjusting for other factors, income was negatively associated with the number of tick submissions. Median household income at the ZCTA level in Massachusetts outside of Boston is relatively high with only 15 ZCTAs having a median household income of less than $35,000, according to the 2018 ACS. As there is a fee associated with submission, we would hypothesize that an analysis at the household level would find household income associated with submission. Further research is needed to look at how to overcome this barrier, such as working with public health departments to subsidize at-risk underserved areas or even insurance coverage of testing fees to allow for accurate, earlier exposure risk assessments for different areas. There may also be opportunity costs in underserved areas that affect submission rates that would need to be addressed. 

### 4.3. Study Strengths and Weaknesses

A major strength of this study was the use of a large data set with minimal missing data. Almost 14,000 human-biting ticks were submitted over a five-year period, allowing an examination of variability in percent positive and range for all four pathogens, including the more rare but emerging *B. miyamotoi*. Study limitations were that TickReport is a passive surveillance system, so there was high variation in tick submissions. The human risk depends not just on the pathogen-positive percentage but on the total number of ticks in an area as well, which is outside this data set. The demographic data were based at the ZCTA level, so individuals submitting ticks may not match the average demographic characteristics of those residing in a ZCTA.

## 5. Conclusions

Passive surveillance of human-biting ticks is an important part of monitoring tickborne diseases. In Massachusetts, the three pathogens previously measured using this data set increased in human-biting ticks since 2012. As messaging consistent with *B. burgdorferi* transmission risk may not be applicable for the less common tick-borne disease pathogens [56], awareness of the variation of the range of high-risk areas for all four pathogens is important to inform public health messages. The association of submissions with education level and other socioeconomic factors at the ZCTA level should be considered to produce more generalizable passive surveillance data, especially when a fee is involved. Further investigation is needed to see if similar associations persist in citizen science projects. Passive surveillance of human-biting ticks and associated pathogens is important for monitoring tick-borne diseases, detecting areas with potentially high risks, and providing public information. We anticipate that the presented results can provide support for medical, public health, and veterinary professionals to continue surveillance for tick-borne disease pathogens and to include socioeconomic determinants of health.

## Figures and Tables

**Figure 1 ijerph-20-04306-f001:**
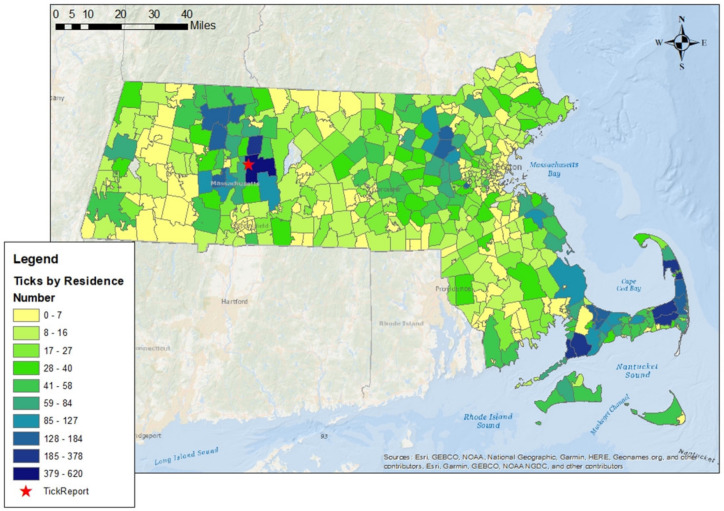
Human-biting *Ixodes scapularis* ticks submitted to TickReport by ZCTA of residence in Massachusetts: 2015–2019.

**Figure 2 ijerph-20-04306-f002:**
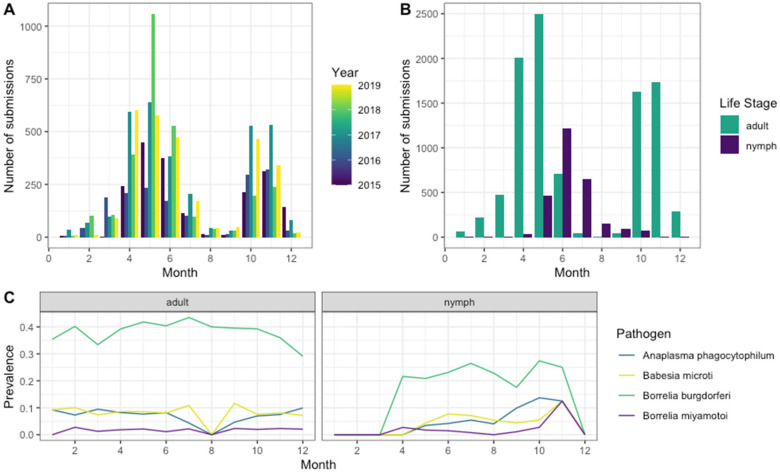
Seasonality of *Ixodes scapularis* submissions by (**A**) Year, (**B**) Life Stage, and (**C**) Prevalence of pathogen-positive ticks by life stage.

**Table 1 ijerph-20-04306-t001:** Annual trends in pathogen-positive *Ixodes scapularis* ticks from exposures in Massachusetts.

	Year	*Borrelia burgdorferi*	*Babesia microti*	*Anaplasma phagocytophilum*	*Borrelia miyamotoi*	Total Number of Ticks
All life stages, including larva (*n* = 228)	Total	35.0%	7.6%	6.9%	1.9%	13,598
Adults	2015	37.98%	6.10%	7.63%	2.40%	1377
	2016	33.95%	7.78%	8.47%	1.85%	1299
	2017	39.60%	8.13%	7.30%	2.18%	2520
	2018	41.37%	8.89%	7.98%	1.47%	2318
	2019	39.34%	8.46%	7.26%	2.19%	2921
	Total	39.00%	8.08%	7.63%	2.01%	10,435
	*p*-value for linear trend	0.48	0.08	0.48	0.82	
Nymphs	2015	19.96%	5.83%	5.16%	0.90%	446
	2016	20.61%	5.45%	3.03%	0.61%	330
	2017	24.96%	6.04%	3.97%	2.07%	629
	2018	24.96%	6.93%	6.20%	1.59%	693
	2019	22.94%	7.05%	5.26%	1.08%	837
	Total	23.13%	6.44%	4.94%	1.33%	2935
	*p*-value for linear trend	0.23	0.08	0.48	0.82	

**Table 2 ijerph-20-04306-t002:** Percent pathogen-positive for nymphs and adults *Ixodes scapularis* by Massachusetts counties (*n* = 13,370).

			Adults					Nymphs		
Counties	Number Submitted	*Borrelia burgdorferi*	*Babesia microti*	*Anaplasma phagocytophilum*	*Borrelia miyamotoi*	Number Submitted	*Borrelia burgdorferi*	*Babesia microti*	*Anaplasma phagocytophilum*	*Borrelia miyamotoi*
Barnstable	2797	37.29%	9.83%	6.51%	2.68%	794	21.41%	9.57%	3.65%	1.76%
Berkshire	429	44.29%	7.69%	12.82%	0.70%	89	24.72%	1.12%	6.74%	0.00%
Bristol	244	33.20%	4.51%	4.92%	2.05%	133	26.32%	5.26%	3.76%	0.75%
Dukes	153	32.03%	6.54%	9.80%	1.96%	204	25.00%	7.84%	8.33%	0.49%
Essex	466	37.55%	10.94%	8.58%	3.00%	110	20.91%	7.27%	5.45%	0.91%
Franklin	1076	43.12%	6.13%	7.34%	0.93%	306	18.95%	1.96%	6.54%	0.65%
Hampden	283	36.40%	6.36%	4.59%	1.77%	44	6.82%	2.27%	13.64%	0.00%
Hampshire	1119	40.66%	5.90%	6.79%	2.41%	270	23.33%	3.70%	2.96%	1.11%
Middlesex	1688	38.57%	8.53%	9.48%	2.19%	382	23.82%	4.97%	4.45%	2.36%
Nantucket	16	56.25%	6.25%	6.25%	0.00%	46	30.43%	6.52%	8.70%	0.00%
Norfolk	478	35.56%	7.32%	6.07%	1.26%	137	27.74%	9.49%	3.65%	0.73%
Plymouth	600	37.50%	9.00%	6.83%	2.00%	233	25.32%	10.30%	5.58%	2.58%
Suffolk	35	31.43%	0.00%	5.71%	2.86%	18	38.89%	0.00%	5.56%	0.00%
Worcester	1051	42.25%	7.52%	8.66%	1.14%	169	26.63%	2.96%	4.73%	0.59%

**Table 3 ijerph-20-04306-t003:** Multivariable negative binomial analysis of the association between socioeconomic factors and the number of *Ixodes scapularis* ticks submitted by Massachusetts residents outside of Boston from 2015–2019 (*n* = 621).

	Final Socioeconomic Model			Final Model with Land Use ^a^			Final Model with Land Use and Amherst Removed ^a^			Final Model with Boston		
	Estimate (SE)	IRR (95% CI)	*p*-Value	Estimate (SE)	IRR(95% CI)	*p*-Value	Estimate (SE)	IRR(95% CI)	*p*-Value	Estimate (SE)	IRR (95% CI)	*p*-Value
Distance to UMass Amherst (km)	0.002 (0.001)	1.002 (1.000–1.004)	0.04 *	0.003 (0.001)	1.003(1.001–1.004)	0.004	0.004 (0.001)	1.004(1.002–1.006)	0.001 **	0.002 (0.001)	1.002(1.0005–1.004)	0.03 *
Log of Population density	−0.19 (0.10)	0.82 (0.64–1.05)	0.05	0.15 (0.12)	1.16 (0.92-1.47)	0.19	0.17 (0.11)	1.18(0.96–1.47)	0.15	−0.40 (0.09)	0.67(0.53–0.84)	<0.001 ***
Percent White (self-reported race)	1.59 (0.55)	4.9 (1.22–18.64)	0.004 **	0.51 (0.56)	1.67(0.82–3.37)	0.36	1.25(0.55)	3.49(1.18–10.26)	0.02 *	1.56 (0.46)	4.77 (1.55–14.2)	<0.001 ***
Percent with a High School Education (or less)	−6.16 (0.47)	0.003 (0.001–0.008)	<0.001 ***	−5.7 (0.43)	0.003(0.001–0.008)	<0.001 ***	−4.88 (0.43)	0.008(0.003–0.018)	<0.001 ***	−5.15 (0.40)	0.006(0.002–0.016)	<0.001 ***
Log of Median Household Income	−1.09 (0.33)	0.34 (0.14–0.78)	0.001 **	−1.01 (0.34)	0.36(0.18–0.71)	0.003 **	−0.46 (0.33)	0.63(0.33–1.21)	0.16	−0.71 (0.31)	0.49(0.21–1.03)	0.02 *
Nagelkerke’s Pseudo-R^2^	27.0%			35.9%			35.7%			NA		

* *p* < 0.05, ** *p* < 0.01, *** *p* < 0.001, a—Adjusted for National Land Cover Database land-use classes.

## Data Availability

The data set is available online (without identifying information) at https://www.tickreport.com/stats.

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
