# Peer review of "Passive Surveillance of Human-Biting Ixodes scapularis Ticks in Massachusetts from 2015–2019"

_ijerph, 2023, doi:10.3390/ijerph20054306_

Round 1
Reviewer 1 Report
For tick submission: It seems white race and higher educated people submitted the most of the ticks.
1. Did author got information on submission related cost? Courier services? That may be a cause for not submitting ticks?
2. Author should provide whether there is any sex bias for submitting ticks
3
Did author find any positive ticks on larvae. E.g., if tick feed by an infected deer mice then that larvae should be positive and that’s one of the way they maintain tick positivity in an ecosystem. They found 228 larve but I did not see any data on B burgdorferi, BAbesia and Anaplasma.
Most of the study finds Nyphal ticks infectivity is higher compare to adults and larvae. Authors of this study found higher ration in adults followed by nymph. What could be the possible explanation?
Did authors capture data on tick attachment time on human? If yeast, It would be nice to see correlation of tick attachment and infection to those 4 agents.
Conclusions should be shortened. It seems very descriptive
Author Response
Dear Reviewer,
Thank you for your time and contribution to this paper. I have addressed each comment below with our answer.
Reviewer 1:
Comments and Suggestions for Authors
For tick submission: It seems white race and higher educated people submitted the most of the ticks.
- Did author got information on submission related cost? Courier services? That may be a cause for not submitting ticks?
We mention in the methods and the discussion that there is a fee associated with submission but we make that more clear by adding the following sentence, “As there is a fee associated with submission, we would hypothesize that an analysis at the household level would find household income associated with submission.”
- Author should provide whether there is any sex bias for submitting ticks
There is not a sex bias, and this has been included in the results section. “Men (n=6743) and women (n=6701) submitted similar number of ticks from exposures (n=154 chose not to include gender).”
- Did author find any positive ticks on larvae. E.g., if tick feed by an infected deer mice then that larvae should be positive and that’s one of the way they maintain tick positivity in an ecosystem. They found 228 larve but I did not see any data on B burgdorferi, BAbesia and Anaplasma.
“Two larva each were positive for B. burgdorferi, A. phagocytophilum, and B. miyamotoi” has been added, thank you
- Most of the study finds Nymphal ticks’ infectivity is higher compare to adults and larvae. Authors of this study found higher ration in adults followed by nymph. What could be the possible explanation?
It is a common finding that adults have a higher percentage positive than nymphs as they have had more blood meals and thus have a higher exposure risk.
See:
Dibernardo, J. K. Koffi, N. H. Ogden and M. A. Kulkarni (2021). "A multi-year assessment of blacklegged tick (Ixodes scapularis) population establishment and Lyme disease risk areas in Ottawa, Canada, 2017-2019." PLOS ONE 16(2): e0246484.
Klitgaard, K., L. J. Kjær, A. Isbrand, M. F. Hansen and R. Bødker (2019). "Multiple infections in questing nymphs and adult female Ixodes ricinus ticks collected in a recreational forest in Denmark." Ticks and Tick-borne Diseases 10(5): 1060-1065.
Rounsville, T. F., G. M. Dill, A. M. Bryant, C. C. Desjardins and J. F. Dill (2021). "Statewide Passive Surveillance of Ixodes scapularis and Associated Pathogens in Maine." Vector-Borne and Zoonotic Diseases 21(6): 406-412.
Wilhelmsson, P., P. Lindblom, L. Fryland, J. Ernerudh, P. Forsberg and P.-E. Lindgren (2013). "Prevalence, Diversity, and Load of Borrelia species in Ticks That Have Fed on Humans in Regions of Sweden and Åland Islands, Finland with Different Lyme Borreliosis Incidences." PLOS ONE 8(11): e81433.
- Did authors capture data on tick attachment time on human? If yeast, It would be nice to see correlation of tick attachment and infection to those 4 agents.
Attachment time was outside of this analysis, and we agree that that would be interesting for future analysis.
- Conclusions should be shortened. It seems very descriptive
The conclusion paragraph has been shortened, thank you.
Reviewer 2 Report
Thank you for your service in providing pathogen testing on submitted ticks as well as highlighting how socioeconomic factors and your fee-based service impacted your submission. Pathogen surveillance is indeed needed and can inform public health messaging and raise awareness for both the general population and medical professionals.
Introduction:
Additional background information is needed in the introduction addressing the basic biology and ecology of these human and (animal?) pathogens and the tick vector. Known risk associated with seasonal, occupational, and other factors previously published for encountering this tick/life stage and pathogens. Does B. burgdorferi s.l. cause Lyme Disease or does this just indicate a complex of species? Which members of this complex are known to cause human Lyme Disease, please address this accurately. Also please address what is not known regarding Bb sl.
Methods:
You only tested adults and nymphs for pathogens but not for the larvae submitted. Can you address the reasons for why larvae were not tested for any pathogens particularly for readers who may not be familiar with the pathogens tested and which pathogens can and cannot be passed via transovarial transmission to larvae. For Ixodes scapularis larvae it would seem reasonable to test generally for Babesia spp. and B. miyamotoi? (also limitations)
You should indicate here why you did not proceed with additional testing on your B. burgdorferi s.l, positives and reference what additional testing would be appropriate if you were to do so.
Results:
It would be useful for the scientific community, public, and medical professionals if the data, either in a table or visual graphic, was presented the submissions by year/month for each life stage and if not too difficult pathogen prevalence. This would add addional data for discussion. Any changes in seasonal life stage submissions/pathogen prevelance compared to previous years? Any times of higher risk?
Discussion:
Please discuss what your B. borgderferi s.l positve results means in context of the Bb sl complex, and comments/requests above.
Any ideas on how you the passive surveillance/fee based testing can be improved to address the disparities in tick submissions you observed.
Author Response
Dear Reviewer,
Thank you for your time and contribution to this paper. I have addressed each comment below with our answer.
Thank you for your service in providing pathogen testing on submitted ticks as well as highlighting how socioeconomic factors and your fee-based service impacted your submission. Pathogen surveillance is indeed needed and can inform public health messaging and raise awareness for both the general population and medical professionals.
Introduction:
Additional background information is needed in the introduction addressing the basic biology and ecology of these human and (animal?) pathogens and the tick vector. Known risk associated with seasonal, occupational, and other factors previously published for encountering this tick/life stage and pathogens. Does B. burgdorferi s.l. cause Lyme Disease or does this just indicate a complex of species? Which members of this complex are known to cause human Lyme Disease, please address this accurately. Also please address what is not known regarding Bb sl.
Information has been added to the introduction as requested, thank you.
The methods used in this study were developed to detect presence of the full range of Lyme borreliosis agents in ticks. However, in North America (or more specifically in Massachusetts), the only agent of Lyme disease is B. burgdorferi sensu stricto. This point has been clarified in the Materials and Methods section.
Methods:
You only tested adults and nymphs for pathogens but not for the larvae submitted. Can you address the reasons for why larvae were not tested for any pathogens particularly for readers who may not be familiar with the pathogens tested and which pathogens can and cannot be passed via transovarial transmission to larvae. For Ixodes scapularis larvae it would seem reasonable to test generally for Babesia spp. and B. miyamotoi? (also limitations)
Larvae were tested and the results have been added.
You should indicate here why you did not proceed with additional testing on your B. burgdorferi s.l, positives and reference what additional testing would be appropriate if you were to do so.
As mentioned above, there no B. burgdorferi sensu lato types other than B. burgdorferi sensu stricto in this region.
Results:
- It would be useful for the scientific community, public, and medical professionals if the data, either in a table or visual graphic, was presented the submissions by year/month for each life stage and if not too difficult pathogen prevalence. This would add addional data for discussion. Any changes in seasonal life stage submissions/pathogen prevelance compared to previous years? Any times of higher risk?
Figure 2 with all year and month, life stage and month and prevalence by life stage and month has been added. In response to this, information has been added to the introduction, results, and discussion on seasonality.
Discussion:
Please discuss what your B. borgderferi s.l positve results means in context of the Bb sl complex, and comments/requests above.
For all intents and purposes, in this region Borrelia burgdorferi s.l. is monotypic (only strict sense forms are found).
Any ideas on how you the passive surveillance/fee based testing can be improved to address the disparities in tick submissions you observed.
“Further research is needed to look at how to overcome this barrier, such as working with public health departments to subsize at-risk underserved areas or even insurance coverage of testing fees to allow for accurate, earlier exposure risk assessments for different areas. There may also be opportunity costs in underserved areas that effect submission rates that would need to be addressed” has been added, thank you
Reviewer 3 Report
Estimated Authors,
Thank you for your contribution to this journal and congratulations for the work done.
Sack et al. report the occurrence of Ixodes scapularis ticks and some of their transmitted pathogens involved in tick bites from Massachusetts during 2015-2019. The authors provide a general picture of the most probably geographic distribution of the tick vector and the main associated pathogens. Additionally, they investigated how socioeconomic factors could influence tick submissions. One of the most strengths of this work is the great contribution of the citizens in tick submissions, which reflects the generalized awareness about the health risks posed by ticks in this area. Moreover, this apparently denotes a well-structured, solid and coordinated passive surveillance activity.
I am very curious about your activity. This tick-testing service is structured within a surveillance activity in collaboration with medical doctors? It is normally publicized by the media or is wide known by the health services?
Do you know something about the health status related to tick-borne diseases of the submitters? People submit the ticks because they feel sick or because they are worried about the tick issue?
The manuscript is well-designed, conducted and written and critically analysed. I accept this manuscript for its publication after minor revisions.
Below you find some minor comments to improve the manuscript.
Introduction
I think that it would be interesting to add some context about the known epidemiological situation about tick-borne diseases in the state of Massachusetts, maybe focusing on the tick-borne pathogens investigated and showing their incidence in humans.
I can imagine that I. scapularis is the most widespread tick species in the state, but I would like to ask you if you have received other tick species from humans. If that is the case, how often are those tick species in humans and why did you decided to focus only on I. scapularis and not to include the other tick species at least in the descriptive analyses?
Materials and methods section
Line 73-74: “The dataset is available online….”. This sentence is unnecessary here because you have already declared this in the dedicated section about data availability.
Line 81-83: This sentences about the pathogens should be in the introduction section where there is a need for some context about the health issue, as I advised you above.
Results
I advise to standardize the form of reporting the percentages in the main text, preferably rounded off to one decimal point. Moreover, I suggest providing a variability measure (e.g., 95% confidence intervals) in the case of infection prevalence reported in the text.
Line 139-40: Although the infection prevalence in ticks seemed to not follow a linear trend, it seems that the number of submitted ticks does, being more evident for nymphs (but also in adults) with a general increase over time. I think that it would be worthy to mention and provide some explanations in the discussion section (e.g., particular increase in temperature in the tick season during the study period? Greater information/publicity provide to the public about the health issue and your services?).
Table 1: I would revise the numbers of ticks reported. According to this table, the total of submitted ticks is 13,370 (10,435+2935), however you report that the total number of ticks is 15,598. What about the remaining 228 ticks?
Line 141-45: The sentences reporting the mean prevalence of B. burgdorferi in adults is repetitive from the previous paragraph while the infection prevalence for nymphs is different… In the previous paragraph you described an infection prevalence of 23.1, here you declared 39.01%. What is the reason? Then, you do not say anything about the other pathogens investigated but only you address the reader to look at the Table 2.
I advise you to not repeat the results already described and to give the same importance to all the pathogens investigated, although B. burgdorferi is the most prevalent. I think it would be nice and helpful to test differences in the pathogen’s infection at the county and tick stage levels. These differences may rely on ecological factors such as land use (at county level) and tick host abundance (at tick stage level).
Table 2: It needs to be formatted. Please, revise the number of ticks reported if it is needed.
Table 3: In the first row of the columns of the second and third models is indicated a subscript letter ‘a’ but not description is provided. The employment of the p-value has been used in the scientific literature since long time, sometimes in a misleading manner (https://doi.org/10.1098/rsbl.2019.0174). I recommend you report the confidence intervals for all IRR of the table, not only for the last model. The interval corresponds to 95% CI? Please indicate in the table.
L153-54: It would be advisable to report also a term of variability. Please, provide the 95% CI of the estimate.
Discussion
L196: You state that the prevalence of pathogens in ticks is not changing, however you have just compared with a previous study, highlighting that the prevalence of the pathogens investigated has increased… So?
Line 202: This percentage is so interesting for me because is so different from the results I usually observed in Europe. From one of the cited papers, I can see that this percentage have been uncovered also in I. scapularis males. I ignore how I. scapularis males behave, but in Europe the males of Ixodes ricinus (the most widespread tick species and main vector for B. burgdorferi) usually show quite low levels of infection by B. burgdorferi. This is mainly associated with feeding behaviour patterns (they do not feed or they do rarely). Do you have any explanation for this high infections levels?
Line 210: Yes, you did not find B. microti in Suffolk County, but I think this could be due to the low sample size tested compared with other counties.
Line 231-234: This association may reflect that counties with greater high education levels are related to a better economic position, so they can afford to pay for tick testing.
Line 238-39: This corresponds to a result and it is not necessary to report here, it is repetitive. Please reformulate the sentence.
Line 250: A reference is required.
Author Response
Dear Reviewer,
Thank you for your time and contribution to this paper. I have addressed each comment below with our answer.
I am very curious about your activity. This tick-testing service is structured within a surveillance activity in collaboration with medical doctors? It is normally publicized by the media or is wide known by the health services?
The activity is primarily with the public and the link to the laboratory is provided though the MA Department of Public Health (https://www.mass.gov/service-details/tick-identification-and-testing-services)
Do you know something about the health status related to tick-borne diseases of the submitters? People submit the ticks because they feel sick or because they are worried about the tick issue?
People may submit ticks for a variety of reasons and that is not information that we collect.
The manuscript is well-designed, conducted and written and critically analysed. I accept this manuscript for its publication after minor revisions.
Below you find some minor comments to improve the manuscript.
Introduction
I think that it would be interesting to add some context about the known epidemiological situation about tick-borne diseases in the state of Massachusetts, maybe focusing on the tick-borne pathogens investigated and showing their incidence in humans.
A paragraph has been added to the introduction, thank you for the suggestion.
I can imagine that I. scapularis is the most widespread tick species in the state, but I would like to ask you if you have received other tick species from humans. If that is the case, how often are those tick species in humans and why did you decided to focus only on I. scapularis and not to include the other tick species at least in the descriptive analyses?
Other species were submitted but we focused on I. scapularis as the one that carried the three most common diseases. This has been added to the methods section and is mentioned in the introduction.
Materials and methods section
Line 73-74: “The dataset is available online….”. This sentence is unnecessary here because you have already declared this in the dedicated section about data availability.
That has been removed, thank you
Line 81-83: This sentences about the pathogens should be in the introduction section where there is a need for some context about the health issue, as I advised you above.
That has been moved, thank you
Results
I advise to standardize the form of reporting the percentages in the main text, preferably rounded off to one decimal point. Moreover, I suggest providing a variability measure (e.g., 95% confidence intervals) in the case of infection prevalence reported in the text.
That has been fixed thank, you and a 95 CI has been added for the infection prevalence
Line 139-40: Although the infection prevalence in ticks seemed to not follow a linear trend, it seems that the number of submitted ticks does, being more evident for nymphs (but also in adults) with a general increase over time. I think that it would be worthy to mention and provide some explanations in the discussion section (e.g., particular increase in temperature in the tick season during the study period? Greater information/publicity provide to the public about the health issue and your services?).
“While not significant, the number of ticks did increase across the study period, this most likely is due to increased awareness of services and not necessarily an increase in tick exposures” has been added.
Table 1: I would revise the numbers of ticks reported. According to this table, the total of submitted ticks is 13,370 (10,435+2935), however you report that the total number of ticks is 15,598. What about the remaining 228 ticks?
The 228 ticks are the larva. They are mentioned in the results section and have been added to the table
Line 141-45: The sentences reporting the mean prevalence of B. burgdorferi in adults is repetitive from the previous paragraph while the infection prevalence for nymphs is different… In the previous paragraph you described an infection prevalence of 23.1, here you declared 39.01%. What is the reason? Then, you do not say anything about the other pathogens investigated but only you address the reader to look at the Table 2.
Thank you for noticing the error that the value for one county and the repetition has been removed.
I advise you to not repeat the results already described and to give the same importance to all the pathogens investigated, although B. burgdorferi is the most prevalent. I think it would be nice and helpful to test differences in the pathogen’s infection at the county and tick stage levels. These differences may rely on ecological factors such as land use (at county level) and tick host abundance (at tick stage level).
We agree that land use by county and tick abundance would be interesting but is outside the focus of this paper. The other pathogens have been added and the paragraph made more comparative and less just the chart.
Table 2: It needs to be formatted. Please, revise the number of ticks reported if it is needed.
Table has been reformatted and the total number of ticks is correct ((n=13370) since the 228 larva are not included in this chart).
Table 3: In the first row of the columns of the second and third models is indicated a subscript letter ‘a’ but not description is provided. The employment of the p-value has been used in the scientific literature since long time, sometimes in a misleading manner (https://doi.org/10.1098/rsbl.2019.0174). I recommend you report the confidence intervals for all IRR of the table, not only for the last model. The interval corresponds to 95% CI? Please indicate in the table.
The 95% CI has been added to the entire chart, thank you
L153-54: It would be advisable to report also a term of variability. Please, provide the 95% CI of the estimate.
The 95% CI has been added, thank you
Discussion
L196: You state that the prevalence of pathogens in ticks is not changing, however you have just compared with a previous study, highlighting that the prevalence of the pathogens investigated has increased… So?
I edited the sentence to make it clearer. The general population of ticks may not be increasing in prevalence but people only encounter a subset of these ticks so people may be encountering more of the positive ones even if the total prevalence does not change. We only tested tick exposures.
Line 202: This percentage is so interesting for me because is so different from the results I usually observed in Europe. From one of the cited papers, I can see that this percentage have been uncovered also in I. scapularis males. I ignore how I. scapularis males behave, but in Europe the males of Ixodes ricinus (the most widespread tick species and main vector for B. burgdorferi) usually show quite low levels of infection by B. burgdorferi. This is mainly associated with feeding behaviour patterns (they do not feed or they do rarely). Do you have any explanation for this high infections levels?
These infection rates are typical of I. scapularis in endemic sites in Northeastern US. While it is true that adult male I. scapularis do not feed on hosts, these ticks acquire infection in pre-imaginal stage bloodmeals (as larvae or nymphs) before becoming adults.
Line 210: Yes, you did not find B. microti in Suffolk County, but I think this could be due to the low sample size tested compared with other counties.
This has been noted, thank you
Line 231-234: This association may reflect that counties with greater high education levels are related to a better economic position, so they can afford to pay for tick testing.
This association continued even after adjusting for household income level.
Line 238-39: This corresponds to a result and it is not necessary to report here, it is repetitive. Please reformulate the sentence.
The sentence has been fixed, thank you
Line 250: A reference is required
It is from the 2018 American Community Survey, which has been added
Reviewer 4 Report
My review is in the file below.

Author Response
Dear Reviewer,
Thank you for your time and contribution to this paper. I have addressed each comment below with our answer.
Review of an article entitled: “Passive Surveillance of Human-biting Ixodes scapularis tick in Massachusetts from 2015-2019.” [International Journal of Environmental Research and Public Health]
Manuscript ID: ijerph-2213545
Authors Sack et al. performed a molecular analysis of human-biting Ixodes scapularis ticks submitted to TickReport tick testing service from 2015-2019 in Massachusetts. Four tick-borne pathogens: Borrelia burgdorferi, Anaplasma phagocytophilum, Babesia microti, and Borrelia miyamotoi were determined by Massachusetts county and by year. Although the results might be important and the total number of ticks tested is impressive (13,598), I found that the manuscript has some important shortcomings as it is, especially in the materials and methods chapter.
I present my questions and comments below:
- Lines 65-66: “Information about the biting tick's species, pathogen species, and tick's feeding status were ascertained by an expert.” - What was the degree of engorgement of ticks. What percentage of ticks were non-engorged, semi-engorged and fullyengorged?
Feeding status was inconsistently reported so it has been removed as one of the information collected
- Lines 66-68: “The Ixodes scapularis genus and species-level identification of each tick was first determined by morphological characterization, then confirmed by a molecular assay.” - By what key or author?
The information has been added, thank you
- Lines 77-78: “. Borrelia burgdorferi s. l., Borrelia miyamotoi, Borrelia mayonii, Babesia microti, and Anaplasma phagocytophilum were detected by a multiplex TaqMan PCR assay.” - The most important reaction conditions and equipment should be described.
The information has been added, thank you
- Lines 122-124: “A total of 13,598 Ixodes scapularis ticks were submitted to TickReport with a reported Massachusetts exposure: 76.73% of ticks were adults (n=10,435), 21.58% were nymphs 124 (n=2,935) and 1.67% were larva (n=228).- It is necessary to specify how many people the ticks were removed from. Was it one tick/one human or otherwise?
The information has been added, thank you. It was one tick per a submission so each one is treated as a separate exposure.
- Tables need better formatting and placement on the page.
Tables have been fixed, thank you
- Discussion - Undergrowths are not necessary here.
I’m sorry but I do not understand this comment.
- Lines 275-277: “We anticipate that the presented results can provide support for medical, public health, and veterinary professionals to continue surveillance for tickborne disease pathogens and to include socioeconomic determinants of health [38,39].” - Here should be found the Authors' own conclusions. Why references are provided here?
References have been removed, thank you
- References: The entire list of references should be provided, there are numerous errors in abbreviations, capital and small letters, etc.
The capitalization and abbreviations have been fixed and all references are provided, thank you
- Only the first time you should use whole Latin names, then abbreviations.
That have been fixed, thank you
Round 2
Reviewer 3 Report
Dear Authors,
Thanks for considering my suggestions. I accept this manuscript for its publication after minor revisions.
Below you find some minor comments to improve the manuscript.
Materials and methods
L84-97: This paragraph is a little confused and should be ordered in a logic way by:
1) Explaining first what Tick Report is,
2) how is the tick submission process
3) giving the reasons why you focus on I. scapularis and Massachusetts and finally
4) concentrate on lab work with the tick identification that will link to the next paragraph.
Moreover, some terms are incorrectly used such as “I. scapularis genus” and some references are missing.
I would suggest modifying this paragraph as follows: “Tick Report is a public outreach service at the University of Massachusetts at Amherst, providing individuals with information about potential pathogen exposures associated with tick bites. The ticks employed for this study were submitted on a voluntary basis to Tick Report from January 2015 through December 2019. All submitters were asked to provide information about the presumed exposure location, the tick removal date, and the person’s sex, age and residence location. Each submission corresponded with a single tick and was treated as a separate exposure. This service is subjected to a fee and is available for the entire United States. Despite the variety of tick species submitted and the vast territory covered by this service, the present work only focused on I. scapularis ticks received from Massachusetts area.
Information about the biting tick’s species and transmitted pathogens were ascertained by an expert [13,18]. Ticks were first morphologically identified to stage and species levels [33-35], then confirmed by molecular assays targeting XXX and XXX genes[?]”.
L109-110: Please, move the reference in brackets to the Reference section.
L110 -112: I think the term “[by us]” can be omitted since you have already described the PCR protocol above, otherwise you should cite a reference. Anyway, this sentence can be deleted it since it does not add more information than that already provided.
Line 114-116: I would rephrase the sentence by adding also that you calculated the 95% CI of prevalence estimates.
Line 153: Please add the comma of the thousands
Line 164: Please, at least for the first 95%CI, at this term: 39.0% (95%CI = 38.1-39.9)
Line 176: What do you mean with ‘HBTs’? There’s not a definition for this term along the text.
Table 3: NCLD? What is it?
Author Response
Dear Reviewer,
Thank you for your continued comments. Sincerely, Lexi
Dear Authors,
Thanks for considering my suggestions. I accept this manuscript for its publication after minor revisions.
Below you find some minor comments to improve the manuscript.
Materials and methods
L84-97: This paragraph is a little confused and should be ordered in a logic way by:
1) Explaining first what Tick Report is,
2) how is the tick submission process
3) giving the reasons why you focus on I. scapularis and Massachusetts and finally
4) concentrate on lab work with the tick identification that will link to the next paragraph.
Moreover, some terms are incorrectly used such as “I. scapularis genus” and some references are missing.
I would suggest modifying this paragraph as follows: “Tick Report is a public outreach service at the University of Massachusetts at Amherst, providing individuals with information about potential pathogen exposures associated with tick bites. The ticks employed for this study were submitted on a voluntary basis to Tick Report from January 2015 through December 2019. All submitters were asked to provide information about the presumed exposure location, the tick removal date, and the person’s sex, age and residence location. Each submission corresponded with a single tick and was treated as a separate exposure. This service is subjected to a fee and is available for the entire United States. Despite the variety of tick species submitted and the vast territory covered by this service, the present work only focused on I. scapularis ticks received from Massachusetts area.
Information about the biting tick’s species and transmitted pathogens were ascertained by an expert [13,18]. Ticks were first morphologically identified to stage and species levels [33-35], then confirmed by molecular assays targeting XXX and XXX genes[?]”.
This has been changed and a sentence on the history of TickReport has been added to the introduction. Thank you for your work and suggestions.
“Tick Report is a public outreach service at the University of Massachusetts at Amherst, providing individuals with information about potential pathogen exposures associated with tick bites. The ticks employed for this study were submitted on a voluntary basis to Tick Report from January 2015 through December 2019. All submitters were asked to provide information about the presumed exposure location, the tick removal date, and the person’s sex, age and residence location. Each submission corresponded with a single tick and was treated as a separate exposure. This service is subjected to a fee and is available for the entire United States. While there was a variety of tick species submitted and states covered by this service, the present work was focused on I. scapularis ticks received from Massachusetts area due to having the most complete information temporally and geographically. Information about the biting tick’s species and transmitted pathogens were ascertained by an expert [13,18]. Ticks were first morphologically identified to stage and species levels [33-35], then confirmed by molecular assays targeting the tick mitochondrial 16S rRNA gene and ITS gene; see reference for a list of primers [13, 18].
L109-110: Please, move the reference in brackets to the Reference section.
This has been fixed, thank you
L110 -112: I think the term “[by us]” can be omitted since you have already described the PCR protocol above, otherwise you should cite a reference. Anyway, this sentence can be deleted it since it does not add more information than that already provided.
The sentence has been removed and a citation added above, thank you
Line 114-116: I would rephrase the sentence by adding also that you calculated the 95% CI of prevalence estimates.
This has been added, thank you
Line 153: Please add the comma of the thousands
This has been added, thank you
Line 164: Please, at least for the first 95%CI, at this term: 39.0% (95%CI = 38.1-39.9)
This has been added to all confidence intervals, thank you
Line 176: What do you mean with ‘HBTs’? There’s not a definition for this term along the text.
That has been removed, thank you. HBT is “human biting tick.”
Table 3: NCLD? What is it?
This has been fixed, thank you